# MACC1 as a Potential Target for the Treatment and Prevention of Breast Cancer

**DOI:** 10.3390/biology12030455

**Published:** 2023-03-16

**Authors:** Mengmeng Lv, Yunjuan Jiao, Bowen Yang, Mengchen Ye, Wenyu Di, Wei Su, Jiateng Zhong

**Affiliations:** 1Department of Pathology, The First Affiliated Hospital of Xinxiang Medical University, Xinxiang 453000, China; 2Department of Pathology, School of Basic Medical Sciences, Xinxiang Medical University, Xinxiang 453000, China

**Keywords:** MACC1, breast cancer, tumor microenvironment, immune escape, radiation resistance

## Abstract

**Simple Summary:**

Despite significant advances in treatment, breast cancer continues to be prevalent around the world. Metastasis associated in colon cancer 1 (MACC1) has been shown to be involved in the progression of more than 20 different types of cancer, including breast cancer, and performs specific functions in different cancers. MACC1’s biological role, molecular mechanism, and pathway are important for understanding cancer progression. MACC1 is highly expressed in breast cancer and has been linked to metastasis, staging, and a poor prognosis. This review article attempts to elucidate the prognostic value of MACC1 expression in breast cancer, its role in immune and anti-radiation therapy, and the network regulatory mechanisms involved, and discusses the application of MACC1 in the treatment and prevention of breast cancer. It is hoped that the use of MACC1 as a biomarker in breast cancer surveillance and clinical guidance can be promoted.

**Abstract:**

Metastasis associated in colon cancer 1 (MACC1) is an oncogene first identified in colon cancer. MACC1 has been identified in more than 20 different types of solid cancers. It is a key prognostic biomarker in clinical practice and is involved in recurrence, metastasis, and survival in many types of human cancers. MACC1 is significantly associated with the primary tumor, lymph node metastasis, distant metastasis classification, and clinical staging in patients with breast cancer (BC), and MACC1 overexpression is associated with reduced recurrence-free survival (RFS) and worse overall survival (OS) in patients. In addition, MACC1 is involved in BC progression in multiple ways. MACC1 promotes the immune escape of BC cells by affecting the infiltration of immune cells in the tumor microenvironment. Since the FGD5AS1/miR-497/MACC1 axis inhibits the apoptotic pathway in radiation-resistant BC tissues and cell lines, the MACC1 gene may play an important role in BC resistance to radiation. Since MACC1 is involved in numerous biological processes inside and outside BC cells, it is a key player in the tumor microenvironment. Focusing on MACC1, this article briefly discusses its biological effects, emphasizes its molecular mechanisms and pathways of action, and describes its use in the treatment and prevention of breast cancer.

## 1. Introduction

The International Center for Research on Cancer reported that there were 2.26 million newly diagnosed cases of BC in the world in 2020 [1]. The second-leading cause of cancer-related death in women is BC, which is the most often diagnosed malignancy in women [2]. BC is a complex illness. In order to differentiate prognosis and direct treatment, the molecular classification of BC is based on immunohistochemistry (IHC) subtypes such as the expression status of the estrogen receptor (ER), progesterone receptor (PR), and human epidermal growth factor receptor-2 (HER2), and as a result, its pathological characteristics and treatment vary significantly [3]. Since many of the BC subtypes represent biologically distinct disease entities, each subtype should have a unique biomarker because each subtype is thought to have a varied prognosis and response to treatment [4,5]. Therefore, the identification of biomarkers is crucial for directing clinical action and patient classification in the direction of personalized medicine.

It is important to select the most appropriate adjuvant treatment for each patient with early BC, identifying those at higher risk of recurrence, but it is also vital to avoid overtreatment of those at low risk of recurrence. Early BC is difficult to detect and intervene in time [6]. Advanced BC patients’ prognosis impacted by the likelihood of recurrence, metastasis, and medication resistance. The main causes of death for BC patients are recurrence and metastasis, which poses a serious threat to patients’ lives and wellbeing. Therefore, to increase the survival of BC patients, it is crucial to discover biomarkers that can recognize and evaluate prognosis early. A good biomarker should be able to predict prognosis in addition to therapy response [7]. The formation of distant metastases is a decisive and deadly event in the course of the disease and is the most common cause of treatment failure [8]. Colorectal oncogenes called MACC1 are linked to metastasis. More than 20 solid cancer types, including BC, have MACC1 identified as a crucial actor and biomarker for tumor growth and metastasis [9,10,11].

## 2. MACC1

### 2.1. MACC1 Biology

By employing human colon cancer tissue, metastases, and normal colonic mucosa in differentially expressing RT-PCR, Stein et al., in 2009, discovered and cloned full-length cDNA for MACC1, a differentially expressed complementary DNA (cDNA) fragment [12]. According to the study, MACC1 is significantly linked to the development, invasion, and metastasis of numerous malignant cancers. MACC1 is involved in promoting epithelial mesenchymal transformation [13], in cancer metastasis [14,15], in blood vessels [16] and lymphatic vessels [17], and in enhancing the Warburg effect [18,19] and cancer immunotherapy [20]. MACC1 is a potentially effective therapeutic target for solid tumors’ anti-tumor and anti-metastatic intervention techniques.

On human chromosome 7 (7p21.1), the human MACC1 gene is roughly 82.7 kb long, and has seven exons and six introns (Figure 1). LNCRNA MACC1-AS1 is a homologous antisense RNA of the last intron of MACC1 mRNA [21]. Hepatocyte growth factor (HGF) and its receptor c-Met, which constitute the primary signaling pathway for MACC1, are also present on chromosomes 7p21 and the vicinity, respectively. Additional tumorigenic and metastasis-related genes are also present in this region (7q21.1 and 7q31.2) [22]. Aberrations of chromosome 7 may represent a mechanism of activation of MACC1 expression.

The cDNA encoding MACC1 contains 2559 nucleotide sequences encoding 852 amino acids [12]. The MACC1 protein structure conserved and consists of four domains: ZU5, Src-homolo-gy 3 (SH3), and two C-terminal death domains (DD) (Figure 1). The rather unstructured N-terminus of MACC1 contains similar interaction motifs for clathrin-mediated endocytosis (clathrin box, NPF, DPF) as seen in the MACC1 homolog SH3BP4/TTP. The ZU5 domain, which is at the N-terminus, is made up of two plates sandwiched together into a common sandwich structure. It is primarily concerned with the adhesion of contractile proteins, and ZU5-DD is engaged in controlling the apoptotic process [23]. Proline-rich motifs ((PxPxP, KxxPxP)) and SH3 domains play a part in interactions between proteins [10]. The Src homologous domain’s SH3 domain, which is upstream of the DD domain and strongly binds to the receptor tyrosine kinase’s phosphorylation site to facilitate protein–protein interaction, is implicated in signal transduction, cell proliferation, and metastasis [24]. The biological function of MACC1 depends on the SH3 domain; without the SH3 domain, MACC1 cannot move from the cytosol to the nucleus and its capacity to trigger c-Met transcription is inactivated [12]. There are two death domain family (DD) domains at the MACC1 N-terminus, one of which is strongly associated with the programmed death of cells and plays a role in the development of inflammation, autoimmune modulation, the directional migration of tumor cells, and other functions [25]. The expression of MACC1 in diverse tumor types and its method of involvement in carcinogenesis and development have been the subject of an increasing number of studies in recent years. All tumor types were found to express MACC1, though to variable degrees. MACC1 was mostly expressed in the small intestine and only weakly expressed in the majority of other organs, including the esophagus, mouth, and colon. Additionally, the kidneys and uterus, which are parts of the reproductive and urinary systems, express MACC1 at relatively high levels. Atypical DNA methylation may play a significant role in the variability in MACC1 expression in various tumor types [11]. The unique structural characteristics of MACC1 help us better understand its mechanism of action in cancer.

### 2.2. Signaling Pathways in Which MACC1 Participates

MACC1′s function in promoting cancer progression stems from its influence on a variety of carcinogenic signaling pathways, such as the HGF/c-Met signaling pathway, Akt signaling pathway, TWIST1/2 signaling pathway, and MAPK signaling pathway, and may be related to the Nanog/Oct-4 pathway and STAT signaling pathway (Figure 2).

When MACC1 was first associated with colorectal cancer, it was discovered that the HGF/c-Met signaling pathway played a role in its ability to promote cancer. By binding to the MET promoter regulator activator protein 1 (SP1) and activator protein 2 (AP2), MACC1 activates the HGF/c-Met signaling pathway. This increases the amount of c-Met proteins, which bind to more HGF. More HGF induces MACC1 to move from the cytoplasm into the nucleus, driving the positive feedback loop of MACC1 and promoting cell growth, epithelial–mesenchymal transformation, angiogenesis, and cell invasion [26]. Stein et al. [12] transfected MACC1 into colon cancer SW480 cells that did not express MACC1, and HGF/c-Met expression increased significantly with the increase in MACC1 expression. The HGF/c-Met signaling pathway plays an important role in tumor proliferation, invasion, and metastasis. The involvement of MACC1 in regulating the HGF/c-Met signaling pathway has been confirmed in cancers such as colon cancer [27], breast cancer, ovarian cancer [28,29], lung cancer [30], liver cancer [31], gastric cancer [32], mesocortical tumor [33], malignant brain tumor [34], and kidney cancer [35]. Data analysis by Jan et al. [24] also showed that MACC1 was significantly associated with the key transcription genes MET, HGF, and the recombinant matrix metal thiobarbituric acid (MP7) in more than half of all cancer types. As an important transcriptional regulator that can enhance the expression of c-Met, MACC1 also plays an important role in the occurrence and development of tumors.

Protein kinase B (PKB), also known as Akt, is a serine protein kinase that plays a key role in transmitting extracellular information to various cellular compartments. These compartments then take part in processes such as apoptosis, protein synthesis, the cell cycle, etc., and are intimately linked to tumor invasion and metastasis [36]. Multiple studies have shown that the tumor process mediated by the AKT signal transduction pathway was regulated by MACC1. In gastric cancer cells, MACC1 moves through the AKT pathway to regulate PDL1 expression and tumor immunity [37], promote the Warburg effect [38], and participate in gastric cancer cell epithelial–mesenchymal transformation [39]. In hepatoma cells, MACC1 was verified as a target of miR-34a and miR-125a-5p. MiR-34a and miR-125a-5p restrained proliferation and metastasis while motivating apoptosis in HCC cells through the PI3K/AKT/mTOR pathway by repressing MACC1 [40]. In cervical cancer, MACC1 induces migration, invasion, stemness, and the inhibition of apoptosis in cervical cancer cells by regulating the AKT pathway [15], and promotes cell invasion and angiogenesis [41]. In osteosarcoma, MACC1 regulates the proliferation, colony formation, invasion ability, cell cycle distribution, apoptosis, and tumorigenicity of human osteosarcoma by altering the Akt signaling pathway [42]. In lung cancer [43] and ovarian cancer [39], overexpressing MACC1 has been demonstrated to overcome chemotherapeutic resistance, and down-regulating MACC1 expression has been found to increase the sensitivity of cisplatin-resistant cancer cells to the drug [44]. MACC1′s involvement in the AKT signaling pathway has also been found in nasopharyngeal carcinoma [45], colon cancer [46], pancreatic cancer [47], and human glioblastoma [48]. The biological role of MACC1 in malignant tumors is related to the state of the Akt signaling pathway.

The TWIST1/2 signaling pathway promotes tumor invasion and metastasis by regulating the transformation of tumor epithelial mesenchymal. According to the research by Wang et al. [49], MACC1 nuclear transfer increases Twist1/2 nuclear expression, which decreases E-cadherin expression by raising VEGFR2 and VE-cadherin expression, and encourages the development of VM. In gastric cancer, MACC1 significantly increases the expression of TWIST1 and induces the tubular formation of human umbilical vein endothelial cells (HUVECs) [49]. One of the most crucial cell-signaling pathways in carcinogenesis is the mitogen-activated protein kinase (MAPK) signaling pathway. In the work by Hua et al., it was found that miR-338-3p mimics greatly reduced the expression of the p38 and ERK1/2 proteins, indicating that miR-338-3p may prevent the MAPK pathway from being expressed in cervical cancer [50]. MACC1 has a comparable effect to miR-338-5p overexpression in suppressing the MAPK pathway, and miR-338-3p partially controls the expression of MACC1 in cervical cancer to influence the pathway’s negative regulation. Lemos et al. found that MACC1′s involvement in tumor progression is associated with the newly discovered MACC1/Nanog/Oct4 signaling pathway. MACC1 regulates the expression of Oct4, Stat3, AKT, and ClclinD1 by regulating the transcription of Nanog, thereby promoting cancer metastasis [51].

Through the transcriptional control of the anti-apoptosis proteins Mcl-1 and Bcl2, signal transduction and activators of transcription (STAT) signaling is a crucial process that controls cell survival and apoptosis [52,53]. After knocking out the MACC1 gene, Radhakrishnan et al. found that cancer cells became more susceptible to Fas-mediated apoptosis as a result of a decrease in the phosphorylation level of STAT, which also caused a decrease in Mcl-1 expression and an increase in Fas expression [54]. It was discovered that MACC1 regulates Fas-mediated apoptosis through STAT1/3–Mcl-1 signaling in solid cancers.

## 3. Effect of MACC1 Overexpression on BC

Huang et al. studied the expression status of MACC1 in human BC for the first time [55]. Using procedures like Western blotting and immunohistochemistry, these can possibly determine the level of MACC1 protein expression in both normal breast tissue and breast carcinoma samples. According to statistical analysis, MACC1 expression was substantially correlated with the main tumor, lymph node metastases, distant metastatic classification, and clinical staging in BC patients but not with age, ER and PR status, or HER2 status. Additionally, there was a strong correlation between MACC1 expression and the clinical and TNM stages of BC. Relapse-free survival (RFS) and overall survival are independent prognostic indicators, according to multivariate analysis of the Cox proportional hazards model. MACC1 has a predictive impact on both ER-negative and ER-positive individuals, according to a stratification of BC patients based on ER status. These results imply that MACC1 may function as a biomarker for BC patients’ prognosis and may contribute to the development of BC.

Key regulators of lymph angiogenesis and vascular growth factor (VEGF)-C and VEGF-D are these molecules. In many malignancies, including BC, increased VEGF expression in tumor tissues is strongly correlated with microvascular density and a bad prognosis [56,57]. For instance, miR-338-3p can promote angiogenesis in hepatocellular carcinoma (HCC) by targeting MACC1, b-catenin, and VEGF; MACC1 is positively correlated with VEGF in cholangiocarcinoma [16] and angiogenesis in cervical cancer [41]. MACC1 upregulates VEGF-C/VEGF-D secretion to promote human gastric cancer lymphatic angiogenesis through c-Met signaling. By using immunohistochemistry and real-time PCR, Söyleyici et al. [58] investigated the expression of MACC1 in 66 patients receiving radical mastectomy for invasive ductal cancer and 25 healthy controls undergoing mammoplasty. MACC1 protein and mRNA expression is higher in BC tissues than in normal breast tissues, and it is significantly correlated with clinicopathological prognostic factors such as high histological grade, ER negativity, and HER2 positivity. It is possible that MACC1 does not alter angiogenesis in BC or that MACC1′s proangiogenic role in BC is considerably independent of VEGF, because there was no obvious link between the mRNA and protein expression of MACC1 and the production of VEGF in BC tissue.

Studies have demonstrated that many tumor patients have c-MET overexpression and gene amplification during the incidence and metastasis of their malignancies. C-MET is intimately associated with the occurrence and metastasis of a range of cancers. The MET gene produces the hepatocyte growth factor receptor (HGFR), a tyrosine-kinase-active protein that is linked to numerous oncogene products and regulatory proteins. In vitro, MACC1 has been shown to activate the c-MET’s promoter. Prguda-Mujic et al. examined the probable link between MACC1 and MET expression in BC by analyzing MACC1 expression in 105 primary BC samples using immunohistochemistry and Western blot analysis [59]. Neither the tumor grade, clinical stage, lymph node involvement, PR status, histological type (ductal versus lobular BC), patient age, or overall survival were shown to be substantially correlated with MACC1 expression. However, they were significantly correlated with shorter DFS (disease-free survival) and overall survival as well as the ER status of malignancies. These findings are less consistent with other findings and might be attributed to variations in patient preference, adjuvant therapy, and MACC1 testing methodologies. Additionally, Spearman-Rho analysis performed on MET mRNA expression data collected by microarray analysis, utilizing an Affymetrix HG-U133A chip for the study, revealed no connection between MACC1 protein levels and MET mRNA expression. Therefore, in contrast to in vitro studies, MACC1 may not be employed as a transcription factor to induce c-MET transcription in BC cells in vivo. Similar to the findings of studies on gastric cancer, MACC1 is significantly independent of MET in BC [60].

SNP stands for single nucleotide polymorphism, and it describes the diversity of DNA sequences brought on by changes in single nucleotides at the genomic level. SNPs can cause a variety of human diseases, including cancer, and conditions linked to particular SNPs will probably become important genetic targets for therapeutic procedures. Certain SNPs are involved in the metabolism of drugs, and SNPs are also key to personalized medicine [61]. Comparing several parts of the same genome in the study of genome-wide association is the most crucial step in biomedicine. According to epidemiological research, SNPs in MACC1 may increase a person’s risk of developing cancer. Zheng et al. found that SNP rs1990172 and SNP rs975263 in MACC1 were significantly associated with a recurrence of hepatocellular carcinoma in hereditary liver transplantation patients [62]. SNP rs975263 may be connected to the lower metastasis-free survival rate of young colon cancer patients with an early TC genotype, whereas the rs3735615 C allele has a protective effect on overall survival. Patients with colorectal cancer who have the MACC1 SNP rs1990172 G allele have a significantly lower overall survival rate [63]. MACC1SNP rs4721888 is significantly associated with susceptibility to oral cancer, and SNP rs975263 is significantly associated with the risk of metastasis in oral cancer patients [64]. SNPs may change MACC1 expression or function, thus affecting the prognosis of cancer.

Dai et al. genotyped four single nucleotide polymorphisms (SNPs) in MACC1 (rs975263, rs1990172, rs3735615, rs4721888) [65]. Their relationship with BC risk was investigated in 1143 samples. While rs47218888 enhanced vulnerability to BC, rs975263 was found to have a protective effect on BC risk, which may assist the proper prediction of the clinical course of BC. MACC1 rs1990172 and rs3735615 polymorphisms did not significantly affect BC risk. A stratified analysis revealed that postmenopausal women were more likely to have the variant genotype of rs975263. Additionally, compared to TGGG wild type, the CTGG and CTCG haplotypes were strongly associated with lower vulnerability to BC. Similar research was conducted by Muendlein et al. to examine the effects of the MACC1 polymorphisms rs1990172, rs975263, and rs3735615, but solely in patients with BC that were HER2-positive [66]. In this study, MACC1 rs1990172 but not rs975263 substantially predicted menopausal state. Chemotherapy was substantially correlated with all three SNPs. Other clinical measures, such as tumor size, venous invasion, or ER, PR, and HER-2 levels that were comparable with Dai et al., were not significantly correlated. SNP rs1990172 and SNP rs975263 were substantially related with the risk of BC progression or death, which is in line with the findings of the study on colorectal cancer. Because of their strong correlation with rs1990172, rs975263 and BC prognosis are significantly correlated [65]. Research suggests that the MACC1 protein may be caused by genetic variation, which restricts c-MET expression, hence encouraging the spread of cancer cells. rs3735615 has a protective effect on BC survival. Since C-MET and HER2 work together to promote cell invasion, BCs that express these two receptors may be more aggressive [67].

Despite the fact that several studies have discussed the prognostic value of MACC1 expression in BCs (Table 1), no firm conclusions have been reached because of the contradictory findings caused by the small sample sizes of earlier studies. To ascertain the predictive significance of MACC1 expression in gynecologic and BC, Wang et al. [68] conducted a meta-analysis of 1811 patients included in a comprehensive search of databases such as PubMed, Web of Science, and Embase to determine the prognostic value of MACC1 expression in BC. High expression of MACC1 in BCs predicts shorter OS and RFS, poorer tumor differentiation, more advanced FIGO staging, and earlier lymph node metastasis. MACC1 expression can predict the prognosis of BCs. The results are comprehensive and convincing.

## 4. Value of Serum MACC1 for BC

Circulating biomarkers have been discovered to be accurate indications of the diagnosis, monitoring, and prognosis of a number of cancers through ongoing in-depth research [72,73,74]. The MUC-1 family of mucin glycoproteins (CA 15.3, BR 27.29, MCA, CA 549), carcinoembryonic antigen (CEA), oncoproteins (HER2/c-erbB-2), and cytokeratin (e.g., tissue peptide antigen and tissue peptide-specific antigen), among other serum indicators, have been described for BC. They are not advised for use in clinical practice because of their low sensitivity and specificity. The only markers commonly used are ER, PR, and Her 2. In order to diagnose and predict BC patients, it is crucial to research several markers based on blood proteins and mRNA. It known that MACC-1 expression in BC and outcomes are correlated. The ability of serum MACC-1 levels to serve as BC diagnostic indicators must be determined.

Tan W et al. measured serum MACC1 levels in 378 BC patients, 120 patients with benign breast disease, and 40 healthy volunteers by using ELISA [69]. The diagnostic and prognostic value of preoperative serum MACC1 levels in BC patients was determined. ELISA measurements of serum MACC1 levels in BC patients and normal healthy controls showed that mean serum MACC1 was significantly higher in BC patients than in controls. High levels of serum MACC1 correlate with TNM stage (I, II, or III), lymphatic metastases, tumor size, and the Ki-67 status of BC. MACC1 levels are unaffected by the presence or absence of distant metastases, the presence or absence of ER, PR, or Her-2 receptor status. MACC1, like TNM staging, predicts disease-free survival (DFS) in BC, and ROC analysis results showed that serum MACC1 measurement successfully distinguished BC patients from normal and healthy controls, with an optimal cut-off value of 38.35 pg/mL, sensitivity of 71.4%, specificity of 89.1%, and AUC of 0.766. Higher serum MACC1 levels were linked to worse DFS compared to lower serum MACC1 levels, according to Kaplan–Meier log-rank analysis.

In a related study, Ahmed et al. investigated the relationship between serum MACC-1 levels and clinicopathological characteristics using 80 patients with benign illnesses as the control group and 60 patients with various stages of BC as the experimental group [75]. His study’s findings are in line with those of Tan W et al., which demonstrate that serum MACC-1 might be employed as a possible biomarker for tumor growth and diagnosis in BC patients. The best cut-off value according to ROC analysis was 2.12 ng/mL, with a sensitivity and specificity of 96.7% and 92.5%, respectively, and an AUC of 0.98. This may explain why the samples were different.

The study by Tan W et al. provides the first evidence that serum MACC1 may be the best biomarker for the diagnosis and prognosis of BC. Clinical laboratories can quickly test serum MACC1 thanks to commercially accessible kits [69]. Clinicians may use serum MACC1 as a biomarker to identify BC. The diagnostic potency of serum MACC1 could not be compared with current BC biomarkers because biomarkers including CA153, CEA, and CA125 were not found in the trial control group. Larger multicenter studies are required to determine whether MACC1 can replace biomarkers in other tissue samples to determine the development or progression of BC. Studies have some limitations, such as the relatively small number of patients studied and the fact that the majority of serum MACC1 is derived from tumor tissue. In conclusion, any investigation into serum MACC1 in breast cancer has the potential to uncover fresh information on how highly circulating MACC1 in liquid biopsy (non-invasive) can be used to predict the progression and formation of metastases in breast cancer (e.g., patient blood).

Overall, study results promote the use of MACC1 as biomarker in the monitoring and clinical guidance of breast cancer. In the future, translational research should pay increased attention to MACC1 and its diagnostic potential for BC to improve treatment, which is currently limited, and accelerate the progress of research from bench to bedside.

## 5. MACC1 Enhances Immune Escape in BC

Immunotherapy has grown in importance as a cancer patient treatment choice in recent years. MACC1 and cancer immunology have been linked in numerous studies. Through TIMER and UALCAN, Hu et al. made it known for the first time that MACC1 expression is strongly correlated with genes relevant to immunological checkpoints [11]. MACC1 expression levels were significantly correlated with immune invasion (B cells, CD4+ T cells, CD8+ T cells, neutrophils, macrophages, and dendritic cells) and most immune markers in more than two dozen cancers studied, but the nature of this correlation varied between cancer types. MACC1 expression tightly correlated with several indicators of Treg and T cell depletion, indicating that MACC1 may play a role in immunological evasion. In a variety of tumor forms, programmed cell death ligand 1 (PDL1) is connected to immune evasion. MACC1 regulates PDL1 expression in GC cells via the c-Met/AKT/mTOR pathway, as verified by Tong and his coworkers [37]. For the first time, it was discovered by Xiong et al. [20] that the MACC1 expression level negatively correlates with infiltration levels and several immune checkpoint biomarkers. High MACC1 expression has a lower response rate with ICIs in COAD. MACC1 was substantially related to treatment for BC in Muendlein et al.’s research. MACC1 is a new immunotherapy target that can be exploited as a predictor of the immune response in tumor patients.

BC comes in a variety of biological subgroups. Immune infiltrates were more frequently observed in the highly proliferative subtypes, which are typically the triple negative and HER2-overexpressing BCs. Studies conducted over the years have revealed that BC is not a particularly immunogenic tumor type, and immune escape is a characteristic of BC. Both native and acquired immune cells have been found to significantly penetrate the BC tumor microenvironment [76]. Studies have revealed that MACC1 shields metastasis from immunological disruption by modifying the tumor microenvironment (TME). Different cells positioned in the tumor microenvironment (TME) can promote tumor development and immune evasion [77]. TAMs, or tumor-associated macrophages, are a crucial part of TME. TAMs can develop into anticancer M1 macrophages or pro-tumor M2 macrophages, which impair adaptive immunity and promote tumor growth and angiogenesis [78,79]. For polarized M2 macrophages, CD163 is a monocyte/macrophage marker that is extremely specific. Poor BC prognostic parameters have been demonstrated to be correlated with high CD163+ TAM density [80].

The development of tumors has been shown to be significantly influenced by tumor-infiltrating lymphocytes (TILs), which are composed of monocytes and multinucleated immune cells (T cells, B cells, NK cells, macrophages, neutrophils, DC cells, etc.) and have prognostic and possibly predictive value [81]. Reduced NK cell infiltration is linked to tumor progression and aggressiveness in BC patients [82]; extensive cytotoxic CD8+ T cell tumor invasion is strongly linked to patient survival [83,84] and therapeutic response [85]; and the presence of CD4+ regulatory T cells (Tregs) correlates with both good and bad outcomes [86]. 

The potential relationship between the relative expression of MACC1 mRNA and various immune cells (CD163+ TAM, CD8+ CTL, and CD56+ NK cells) in the tumor microenvironment was investigated in order to investigate whether MACC1 can control the immunological characteristics of cancer cells while enhancing the ability of tumor cells to metastasize. MACC1mRNA expression was found in 120 BCs and paracancerous tissues using quantitative reverse transcription polymerase chain reactions, according to Ali et al. [70]. MACC1mRNA levels were found to be positively correlated with CD163+ tumor-associated macrophages and negatively correlated with CD56+ natural killer cells and CD8+ cytotoxic T cells. It was discovered that the expression of MACC1mRNA in cancer tissues was significantly higher than that in non-cancer tissues. Poor prognostic factors include big tumors, grade III tumors, positive lymph node metastases, lymphovascular invasion, and stage III cancers, and elevated Ki-67 expression was substantially related with high MACC1 expression. OS and PFS are correlated with MACC1 expression. According to multivariate analysis, lymphatic invasion and MACC1mRNA expression might both be employed as independent predictors of BC prognosis. It is clear that MACC1 can influence immune cell infiltration into the tumor microenvironment, facilitate tumor cell immune evasion, and serve as a solid independent prognostic indicator for BC. Additionally, the relative expression of MACC1 mRNA in the molecular subtype showed substantial variations, according to the study. Although it may have been because of the study’s relatively small sample size of TNBC cases, there was no correlation between the relative expression of MACC1 mRNA and the ER, PR, or HER2 status. Finally, given the emergence of checkpoint immunotherapy, targeting MACC1 may help in the construction of new combination ideas for the better immune surveillance of breast cancer. MACC1 not only encourages the spread of metastases but also protects them from immunological disruption in breast cancer.

In conclusion, while studies have shown that MACCI is associated with tumor progression and immune cell infiltration in a variety of cancers, including breast cancer, more efforts must be made to provide reliable data to support their analytical validity and, ultimately, their clinical validity and utility. MACC1 is expected to predict biomarkers for BC immunotherapy in a reliable and exhaustive manner.

## 6. MACC1 Enhances the Radiation Resistance of BC

Surgery, chemotherapy, and radiotherapy are the three primary cancer treatment modalities. Ionizing radiation (IR), which has a high intensity, is used in radiation therapy to treat tumors by eradicating cancer cells. Radiation therapy is a successful cancer treatment, but the majority of patients eventually develop radiation resistance and experience cancer recurrence [87]. Radiation therapy is used in the treatment of more than 50% of cancer patients. In most situations, radiation therapy uses X-ray ionizing radiation to cause DNA damage by directly or indirectly targeting the DNA backbone and inducing a chain reaction of biological reactions. However, due to BC’s innate radioresistance, investigations have indicated that the efficiency of IR therapy is not particularly optimal compared to other malignancies [88]. MACC1 has been linked to BC radiation resistance, according to research by Zhang et al. IR therapy in combination with other modalities has shown promise in the treatment of BC.

Radiation therapy is an integral part of the multidisciplinary management of breast cancer. Radiation therapy can be used to treat almost every stage of breast cancer. However, after radiotherapy, breast cancer cells significantly enhanced their tolerance, invasion and spheroidous ability to radiotherapy. As a result, it is critical to clarify the causes of radiation resistance for breast cancer for the advancement of breast cancer research. MACC1 upregulation has been shown to promote radiotherapy tolerance. Li et al. [89]. studied the exact role and molecular mechanism of the oncogene LncRNA FGD5-AS1 and its associated MACC1 in radiation-resistant breast cancer by establishing a radiation-resistant breast cancer cell line. FGD5-AS1 and MACC1 are highly expressed in cancer tissues and radiation-resistant cell lines, and they improve cancer cell radiation resistance. FGD5-AS1 deletion inhibited the G0/G1 phase of breast cancer cell lines and induced more apoptosis when exposed to 2 Gy X-ray, triggering apoptosis in BC cells. After studying the mechanism of anti-radiation action of FGD5-AS1 in breast cancer, LncRNA FGD5-AS1 was found to facilitate the radioresistance of breast cancer cells by enhancing MACC1 expression through competitively sponging miR-497-5p. After interrupting the function of MACC1, breast cancer cells triggered the apoptosis pathway and regained sensitivity to ionizing radiation. This phenotype could be rescued by exogenous MACC1 expression or miR-497 depletion.

In conclusion, MACC1 may be involved in the radiation sensitivity of breast cancer cells via the LncRNA FGD5-AS1/miR-497-5p/MACC1 axis, and apoptosis pathway blockade is the main mechanism of LncRNA FGD5-AS1 induced radioresistance, providing a new direction of focus for breast cancer radiotherapy. However, the specific regulatory mechanism of MACC1 in radiation-resistant breast cancer requires further investigation.

## 7. Epigenetic Regulation of MACC1 in BC

The development of CRC is characterized by MACC1, which has been linked to numerous molecular pathways. On the other hand, little is understood about MACC1’s molecular mechanism in BC. The first transcriptional target of MACC1 was discovered to be the Met receptor tyrosine kinase gene. The HGF/Met/MACC1 axis plays a well-known role in CRC and CRC transfer. In BC, the disruption of the c-Met pathway has been shown to lead to a worse prognosis [89,90] and lead to resistance to endocrine therapy or trastuzumab therapy [91,92]. HGF, which is a c-ligand, is also regarded as a separate prognostic factor for BC [93]. It is crucial to comprehend the significance of MACC1 as a potential regulator of the BC HGF/c-Met cascade.

In order to determine if MACC1 is a significant prognostic factor for BC, Sueta et al. performed a thorough gene expression investigation of MACC1 using IHC and RT-qPCR in 300 BC patients [71]. MACC1, c-Met, and HGF mRNA expression levels in BC and healthy breast tissue were examined. MCF-7 and T47D, luminal-type cell lines, had decreased MACC1 protein levels compared to HER2 (MDA-MB-453) or triple-negative cell lines (MDA-MB-468). In BC cell lines, MACC1 and c-Met expression is significantly different. In BC cell lines, MACC1 overexpression has little impact on the expression of the c-Met protein. The effect of MACC1 on the biological function of BC was further evaluated. It was discovered that MACC1 did not promote c-Met expression after transfecting BC cell lines, and that MACC1 did not result in substantial changes in migration or the ability to provide value. After HGF induction, there was likewise no discernible difference in the expression of c-Met. MACC1 and the c-Met promoter area in BC were not identified to bind after ChIP analysis utilizing the HaloCHIPTM technique, and the findings were independently confirmed. MACC1 is probably not acting on BC through the HGF/c-Met signaling pathway, in contrast to earlier investigations of MACC1 in CRC. MACC1 and HGF/c-Met signaling have been found to have different regulatory effects on BC. This is because c-Met transcription can also be activated by other factors, including hypoxia-inducible factor 1 and activating protein-1.

## 8. MACC1-AS1 Participates in BC’s ceRNA Regulatory Network

Competitive endogenous RNA (ceRNA) is a transcript that engages in competitive miRNA sharing to regulate itself at the post-transcriptional stage. ceRNA networks connect the function of non-coding RNAs such microRNAs, long noncoding RNAs, pseudogene RNAs, and circular RNAs with the function of protein-coding mRNA. Cancer is just one of the many human diseases that can be brought on by ceRNA network dysregulation [94]. MiRNAs, which are at the heart of ceRNA interactions, have been considered as diagnostic indicators or therapeutic targets for cancer since they are involved in the pathogenesis of a number of human disorders, including cancer. Long noncoding RNA (lncRNA), which takes part in specific gene expression pathways as oncogenes or gene regulators, can facilitate miRNA-dependent gene regulation [95,96].

MACC1-AS1, a homologous antisense lncRNA for the sixth intron of the MACC1 gene, has been linked to the development of certain malignancies through influencing miRNAs, according to numerous studies. MACC1-AS1 promotes proliferation by downregulating PTEN in lung adenocarcinoma cells [97], promoting invasion and proliferation by regulating paired-box gene 8 (PAX8) in hepatoma cells [98], promoting stemness by antagonizing miR-145 [99], and upregulating cyclin-dependent kinase 6 in cervical squamous cell carcinoma [100]. In pancreatic cancer, it promotes cancer progression by activating the PAX8/NOTCH1 signaling pathway [101].

MACC1-AS1 is a cognate antisense lncRNA of the sixth intron of the MACC1 gene. The role of MACC1-as1 in promoting breast cancer progression may be through posttranscriptional regulation of MACC1 mRNA, acting as a cell growth regulator to promote breast tumor progression. Zhang et al. [102] investigated the underlying molecular mechanism that regulates MACC1-AS1-mediated cell proliferation and breast tumor progression by identifying its interacting molecules. The study found a role for MACC1-AS1 in promoting breast cancer cell proliferation by reducing their activity against pleiotrophin and c-Myc mRNA through competitively sponging the corresponding miRNAs, including miR-145-3p and miR-384. PTBP1 is a MACC1-AS1-binding partner. The binding of PTBP1 not only facilitates the stability of MACC1-AS1 and enhances the sponge effect of MACC1-AS1 on miRNAs, but also competes with PTBP1 target mRNAs for PTBP1 binding. miR-145-3p indirectly induces the downregulation of PTBP1 transcription through repressing c-Myc expression. There is a positive feedback loop between MACC1-AS1 and PTBP1 interaction. PTBP1 and miRNA(miR-145-3p and miR-384 coordinate when binding to lncRNA MACC1-AS1, and the balance of this interaction directly affects the breast cancer cell phenotype. The study observed that MACC1AS1-mediated breast cancer progression is associated with an increased expression of MACC1 mRNA. MACC1-AS1 can be a binding platform for target miRNAs and PTBP1, and the binding efficiency eventually modulates the post-transcriptional fate of genes targeted by the miRNAs and PTBP1.

## 9. Conclusions and Future Perspectives

MACC1 increases tumor cell motility and invasion and causes metastasis in solid tumors, despite being primarily engaged in the gene c-MET that directly produces metastasis. However, years of research have revealed a complex and profound association between MACC1 and a number of malignancies, including breast cancer, and that MACC1 plays a more significant and specific function in particular cancers. For early cancer identification and prevention, various routes allow us to undertake genetic screening or testing, such as MACC1 testing in blood, before cancer fully develops. According to the evidence in this review, aberrant MACC1 expression is a reliable indicator of the prognosis of breast cancer, including OS and RFS, tumor differentiation staging, positive lymph node metastases, etc. The majority of studies have shown no significant relationship between MACC1 and the clinical pathological prognostic parameters of breast cancer such as ER, HER2, PR, and Ki-67, and it is necessary to increase the study sample size for in-depth research given the conflicting results between these studies.

MACC1 may also be a promising target for preventing the growth of breast cancer, particularly in individuals with advanced breast cancer, given its involvement in a number of malignant biological activities and its negative impacts on BC therapy. The intricate MACC1 regulatory network, which is made up of numerous signaling pathways, kinases, transcription factors, and miRNAs and participates in mediating the tumor microenvironment, enables a more thorough understanding of BC and allows for the further investigation of the role of MACC1 in the expression/activity of other carcinogens. The effectiveness of MACC1 inhibition may be increased by concurrently inhibiting MACC1 and the regulatory network components that it are linked to it in the tumor microenvironment. MACC1 has also been linked to radiation resistance and breast cancer immunity, shedding more light on breast cancer treatment plans. Due to these factors, we may be able to develop BC treatment methods and further our understanding of breast cancer prevention by targeting MACC1 in these situations. In the future, translational research should focus more on MACC1 and its therapeutic potential for BC in order to expand on the few available treatments and quicken the transition from clinical to therapeutic research.

## Figures and Tables

**Figure 1 biology-12-00455-f001:**
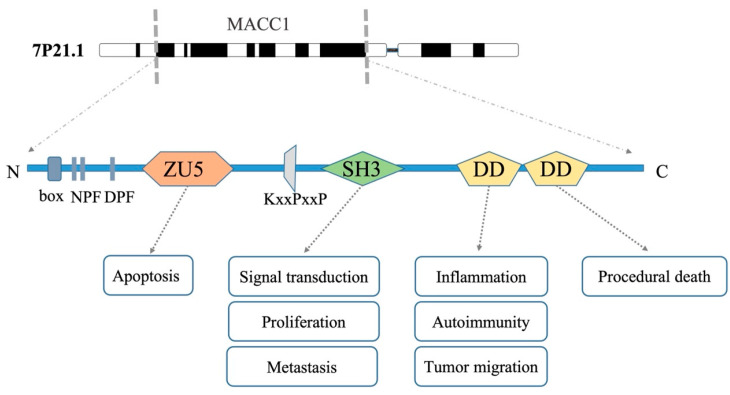
MACC1 domain and potential function. Interaction motifs for clathrin-mediated endocytosis (clathrin box, NPF, DPF); the functions of the ZU5 domain at MACC1 N-terminus are mainly associated with apoptosis; proline-rich motifs (KxxPxP) and SH3 (Src homology domain) domains play a part in interactions between proteins; there are two death domain family (DD) domains at MACC1 C-terminus.

**Figure 2 biology-12-00455-f002:**
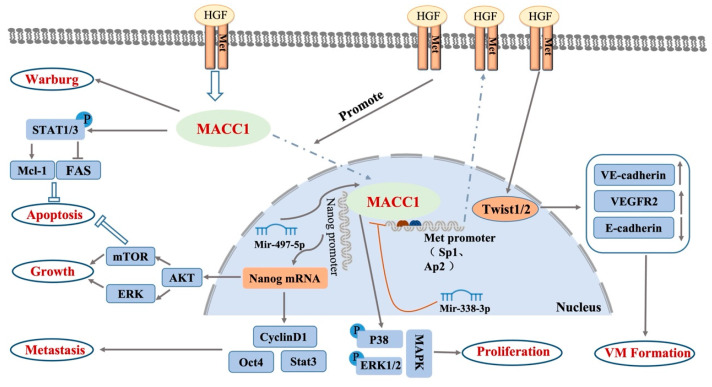
MACC1 effectors, signaling mechanisms, and biological responses. MACC1 is involved in a variety of cancer hallmark functions such as proliferation, metastasis, apoptosis resistance, angiogenesis and Warburg through transcriptional activation of its key target molecules HGF/c-Met, Akt, TWIST1/2, MAPK, Nanog/Oct-4, and STAT.

**Table 1 biology-12-00455-t001:** The correlation of MACC1 expression with clinical features of breast cancer.

Ref.	Histopathology	Staging	ER	HER2	PR	Ki-67	OS	DFS/RFS	Sample Size
[55]	+	+	−	−	−	…	+	+	397
[58]	−	−	+	+	−	−	…	…	91
[59]	−	−	+	…	−	…	−	+	105
[68]	+	+	…	…	…	…	+	+	1811
[69]	+	+	−	−	…	+	…	…	538
[70]	+	+	−	−	−	+	+	+	120
[71]	−	−	−	…	…	…	…	+	300

+, Correlation; −, independent; …,with reported missing data; Staging, TNM or other staging; ER, estrogen receptors; HER2, human epidermal growth factor receptor 2; PR, progesterone receptors; Ki-67, proliferative index; OS, overall survival; DFS/RFS, disease-free survival/ progression-free survival.

## Data Availability

The datasets used during the current study are available from the corresponding author on reasonable request.

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
