# Peer review of "MACC1 as a Potential Target for the Treatment and Prevention of Breast Cancer"

_biology, 2023, doi:10.3390/biology12030455_

Round 1

Reviewer 1 Report (Previous Reviewer 1)

The manuscript should be published in this revised version.

Reviewer 2 Report (Previous Reviewer 2)

The manuscript titled MACC1 as a potential target for the treatment and prevention of breast cancer is a topic of relevance to the research community. The authors have evaluated the current literature constructively and comprehensively in the context of Breast cancer.

This manuscript is a resubmission of an earlier submission. The following is a list of the peer review reports and author responses from that submission.

Round 1

Reviewer 1 Report

comprehensive review about the role of MACC-1 in breast cancer. Minor editing is needed. Repeation in line 308 what is already described in line 303.

Reviewer 2 Report

The manuscript titled MACC1 as a potential target for the treatment and prevention of breast cancer is a topic of relevance to the research community, but the authors have failed to evaluate the current literature constructively and comprehensively in the context of Breast cancer. Simply stating the facts from literature does not add value and the authors must comprehensively analyze and put it in context.  

The manuscript is poorly written and is difficult to comprehend.

To name a few other inconsistencies:

1.       The authors need to be careful in maintaining standard abbreviations, for instance MACC1 is Metastasis associated in colon cancer 1 and not Metastasis-associated colorectal cancer gene-1.

2.       Figure wrongly quoted in text – Line 65

3.       Mistakes in figure legend – Line 72 (what is RASA1?)

Mmisleading information in Figure 2 like AP2 transcription factor for activation of c-Met promoter whereas SP1 and SP2 transcription factors mentioned in Line 135.

4.       Misleading claims/ misinterpretations like “death receptors control apoptosis by linking the MACC1 and STAT signals” in Line 196, whereas the quoted manuscript shows MACC1 controls Apoptosis through STAT signaling.

5.       Misleading claims/ misinterpretations “MACC1 immune cell infiltration is a crucial prerequisite for immune checkpoint inhibitors (ICIs) to respond to tumors” in Line 210;

6.       Misleading claims/ misinterpretations “BC mostly examines tissue-based indicators to assess prognosis and forecast therapeutic response” in Line 321.

7.       Exact repetitions of statements in section 7 first (Line 440-445) and second paragraph (Line 446-454) as well as in third paragraph Line 457-463.

8.       Misleading section title: Section 8 (Line 476): This section only talks about MACC1-AS1 lncRNA.

The manuscript requires native speaker editing after re-structuring the whole manuscript. More than a third of the manuscript discusses just MACC1 biology which is comprehensively covered in already existing reviews. It is highly appreciated if the manuscript is restructured in a way the MACC1 biology is discussed in the context of breast cancer. A brief discussion on the current status of Breast cancer research and the need for robust diagnostic, prognostic factors and therapeutics targets need to be discussed along with how MACC1 serves this purpose.

Reviewer 3 Report

1. Please edit DD domain which seems to present on C-terminal but author saying its present on N-terminal. Please confirm it.

2. In Figure 2. Met promoter protein showing Sp1 and Ap2 but in the manuscript it is saying Sp1 and Sp2. Please confirm it.

3. Please clarify in figure 2, how MACC1 regulating Nanog.

4. How Mir-497-5p regulates MACC1?

5. Once read complete manuscript very carefully and critically. Use scientific word for some terms. Some lines showing contradictions too.

6. Some abbreviations should be write in full foms.

7. In section 2.4, what is 19 macrophages, 19 neutrophils, and 16 dendritic cells?

8. Elaborate section 2.4.

9. There are typographical mistakes and some lines have no meaning.

10. SNPs part should be explained separately in section 3.

11. Remove the repeatative lines.

12. In section 4, one line saying " In BC, serum markers mostly used to track the patients with disease." What does it mean?

13. Some references should be added to last paragraph of Section 5. 

14. Elaborate Section 6.